# Quantum Interior Point Methods: A Review of Developments and an Optimally Scaling Framework

Mohammadhossein Mohammadisiahroudi[1][000−0002−4046−0672], Zeguan Wu[1][0000−0002−5695−7579], Pouya Sampourmahani[1][0000−0002−2292−551X], Adrian Harkness[1][0009−0001−5518−6442], and Tamás Terlaky[1][0000−0003−1953−1971]

Lehigh University, Bethlehem PA, USA mom219@lehigh.edu

**Abstract.** The growing demand for solving large-scale, data-intensive linear and conic optimization problems, particularly in applications such as artificial intelligence and machine learning, has highlighted the limitations of classical interior point methods (IPMs). Despite their favorable polynomial-time convergence, conventional IPMs often suffer from high per-iteration computational costs, especially for dense problem instances. Recent advances in quantum computing, particularly quantum linear system solvers, offer promising avenues to accelerate the most computationally intensive steps of IPMs. However, practical challenges such as quantum error, hardware noise, and sensitivity to poorly conditioned systems remain significant obstacles. In response, a series of Quantum IPMs (QIPMs) have been developed to address these challenges, incorporating techniques such as feasibility maintenance, iterative refinement, and preconditioning. In this work, we review this line of research with a focus on our recent contributions, including a novel almost-exact QIPM framework. This hybrid quantum-classical approach constructs and solves the Newton system entirely on a quantum computer, while performing solution updates classically. Crucially, all matrix-vector operations are executed on quantum hardware, enabling the method to achieve an optimal worst-case scalability w.r.t dimension, surpassing the scalability of existing classical and quantum IPMs.

**Keywords:** Quantum Interior Point Method · Quantum Linear System Algorithm · Iterative Refinement · Preconditioning · Linear Optimization · Conic Optimization.

## 1 Introduction

In this paper, we review recent advances in Quantum Interior Point Methods (QIPMs) for linear optimization (LO) problems. The standard form LO problem is minimizing a linear objective function over a polyhedron, formally defined as

$$
\begin{aligned}
\min_{x \in \mathbb{R}^n} \quad & c^T x \\
\text{s.t.} \quad & Ax = b, \\
& x \geq 0,
\end{aligned} \tag{P}
$$

where $A \in \mathbb{R}^{m \times n}, b \in \mathbb{R}^m$, and $c \in \mathbb{R}^n$. It is well-known that there is a dual problem associated with the primal problem as

$$\max_{(y,s) \in \mathbb{R}^m \times \mathbb{R}^n} \quad b^T y$$
$$\text{s.t. } A^T y + s = c, \tag{D}$$
$$s \geq 0.$$

By the strong duality theorem [1], all optimal solutions, if they exist, belong to the set $\mathcal{PD}^*$, which is defined as

$$\mathcal{PD}^* = \{(x, y, s) \in \mathbb{R}^{n+m+n} : \ Ax = b, \ A^T y + s = c,$$
$$x^T s = 0, \ (x, s) \geq 0\}.$$

Linear optimization plays a foundational role in a broad range of fields, including machine learning, operations research, logistics, and finance. Historically, the Simplex algorithm [2] was among the first prominent methods to solve LO problems. While highly effective in many practical instances, Simplex methods can exhibit exponential-time behavior in the worst case [3]. In contrast, the introduction of Interior Point Methods (IPMs) marked a major breakthrough in optimization. Starting with Karmarkar's projective algorithm [4], IPMs have evolved into the most theoretically efficient class of algorithms for solving LO problems, offering polynomial-time complexity with robust convergence guarantees.

Modern IPMs exploit the geometry of the central path, an analytic trajectory defined by a set of perturbed optimality conditions, which guides iterates toward the optimal solution [1, 5]. When initialized appropriately, IPMs follow this path using Newton's method, requiring approximately $\mathcal{O}(\sqrt{n} \log(1/\epsilon))$ iterations to obtain an $\epsilon$-approximate solution [1]. However, a significant computational bottleneck in IPMs lies in solving the Newton linear system at each iteration. Classical direct methods such as Cholesky factorization incur $\mathcal{O}(n^3)$ complexity, which becomes intractable for large-scale, dense problems. Iterative methods, including conjugate gradient (CG) solvers [6, 7], mitigate this challenge with lower per-iteration costs but at the expense of increased sensitivity to matrix conditioning and convergence accuracy.

To further enhance the scalability of IPMs, several improvements have been introduced. These include partial update techniques and low-rank updates, which reduce the cost of computing Newton directions and yield the best-known classical total complexity of $\mathcal{O}(n^3 L)$ for LO problems [1]. More recently, the incorporation of advanced tools such as fast matrix multiplication, spectral sparsification, and stochastic methods have pushed the complexity to $\mathcal{O}(n^\omega \log(n/\epsilon))$, where $\omega < 2.3729$ is the matrix multiplication exponent [8–10]. Alternatively, first-order methods like the primal-dual hybrid gradient (PDHG) algorithm have demonstrated empirical success in solving large-scale LO problems, although they lack rigorous theoretical complexity bounds [11, 12].

Alongside these classical advances, quantum computing has emerged as a powerful paradigm capable of accelerating various computational tasks. Quan-

tum algorithms such as Shor's factoring algorithm [13] and Grover's search algorithm [14] have showcased the potential of quantum computers to achieve polynomial or even exponential speedups. Of particular interest for optimization is the class of quantum linear system algorithms (QLSAs), pioneered by the Harrow-Hassidim-Lloyd (HHL) algorithm [15]. HHL and its successors [16–18] solve sparse quantum linear systems with exponential speedups under certain assumptions, although they exhibit unfavorable dependence on condition number, sparsity, and required precision.

Motivated by the capabilities of quantum computing, researchers have sought to integrate quantum solvers into classical optimization frameworks. This effort has led to the development of Quantum Interior Point Methods (QIPMs), which aim to exploit quantum acceleration in solving the Newton systems arising in IPMs. Notable contributions include quantum subroutines for the Simplex method [19], QAOA for binary optimization [20], and quantum multiplicative weight update methods for semidefinite optimization [21, 22]. For linear and semidefinite programming, QIPMs have demonstrated potential polynomial speedups in terms of problem dimension [23, 24]. However, these early QIPMs faced substantial challenges. The hybrid nature of QIPMs necessitates the extraction of classical information from quantum states at each iteration, typically via quantum tomography algorithms (QTAs). These steps often introduce significant error and computational overhead, diminishing the overall efficiency of the method.

To overcome these limitations, a series of research efforts has led to the development of improved QIPMs. By incorporating iterative refinement and preconditioning techniques, recent frameworks reduce the impact of quantum errors and ill-conditioning, achieving exponential improvements with respect to precision and condition number compared to earlier quantum methods [25–27]. For instance, Wu et al. [28] introduced a dual logarithmic barrier-based QIPM with improved iteration complexity and memory access efficiency via QRAM.

In this work, we review some of these advancements with a focus on the novel, almost-exact QIPM framework that achieves provable quantum advantage. In our proposed approach, the Newton system is both constructed and solved entirely on a quantum computer, while classical computation is reserved only for solution updates. All matrix-vector products, the most expensive components in classical QIPMs, are offloaded to quantum hardware, reducing total runtime. Our hybrid quantum-classical framework achieves optimal worst-case scaling of $\mathcal{O}(n^2)$ for fully dense linear optimization problems, outperforming both classical IPMs and existing QIPMs in terms of dimensional complexity.

This framework supports inexact quantum operations, such as quantum matrix inversion and matrix-vector/matrix-matrix multiplication, through the use of iterative refinement. Crucially, unlike prior QIPMs, our method eliminates all classical matrix operations, resulting in a total classical arithmetic cost of $\mathcal{O}(n^2 \log(1/\epsilon))$. This asymptotically improves upon previous QIPMs by a factor of $\mathcal{O}(\sqrt{n})$ and offers a provable quantum speedup, as any classical analog would require at least $\mathcal{O}(n^{2.5})$ operations.

The structure of the paper is as follows. In Section 2, we discuss how novel reformulations aid in maintaining feasibility and achieving the best-known iteration complexity for QIPMs. Section 3 reviews how iterative refinement and preconditioning techniques can mitigate the effects of ill-conditioning and enhance the precision of QIPMs. In Section 4, we review recent advancements in Quantum Linear System Algorithms (QLSAs) and Quantum Tomography. Section 5 presents the state-of-the-art QIPM based on a novel Almost-Exact IPM framework that achieves optimal scaling. In Section 6, we explore the applications and implications of recent QIPM advancements in artificial intelligence and machine learning. Finally, Section 7 concludes the paper and outlines directions for future work.

## 2   Inexact Feasible QIPMs

In the general scheme of IPMs, we apply the Newton method to the perturbed optimality conditions iteratively to approach an optimal solution by tracing the so-called central path. There are three reformulations of Newton systems to calculate the Newton direction at each step of IPMs in the classical IPM literature. The prevailing system is the Normal Equation System (NES) defined as

$$AD^2A^T\Delta y = Ax - \beta\mu AS^{-1}e,$$

where $A \in \mathbb{R}^{m \times n}$ is the constraint matrix, $D = \text{diag}(x)^{1/2}\text{diag}(s)^{-1/2}$ is the diagonal scaling matrix, and $\mu = \frac{x^T s}{n}$ is the complementarity measure.

One major issue is that an inexact solution to any traditional Newton systems calculated by a QLSA+QTA subroutine may lead to infeasibility. To properly address this infeasibility, inexact infeasible QIPM (II-QIPM) [25] has been developed which has $\mathcal{O}(n^2 \log(\frac{1}{\epsilon}))$ iteration complexity, where $n$ is the number of variables and $\epsilon$ is the target optimality gap.

To improve this complexity, we propose two inexact feasible QIPMs (IF-QIPMs) using two novel reformulations of Newton systems. First, we use a basis for null-space of $A$, stored in columns of the matrix $V$ to reformulate the Newton system in the Orthogonal Subspaces system (OSS) [29] as

$$\begin{bmatrix} -XA^T & SV \end{bmatrix} \begin{bmatrix} \Delta y \\ \lambda \end{bmatrix} = \beta\mu e - Xs. \tag{OSS}$$

We prove that the inexact solution for OSS solution provides a feasible Newton direction.

In another paper, we propose another system that is a modified version of the NES and it is more adaptable for quantum singular value transformation [26]. We prove the iteration complexity for both IF-QIPMs is $\mathcal{O}(\sqrt{n}\log(\frac{1}{\epsilon}))$ which leads to considerable polynomial speed-up in the complexity of QIPMs.

# 3   Iterative Refinement and Preconditioning

Another challenge in QIPMs is that their complexity has polynomial dependence on $\frac{1}{\epsilon}$ because of the QTA's overhead. This means previous QIPMs are not polynomial time algorithms as one needs to reach $\frac{1}{\epsilon} = \mathcal{O}(2^L)$ to find an exact optimal solution for an LO problem, where $L$ is the binary length of input data. We use an iterative refinement technique to address this issue [26, 29].

Iterative refinement has been widely used in classical numerical algorithms to improve the accuracy of solutions to linear systems. We adapt this technique to iteratively use limited-precision IF-QIPM to obtain a higher-precision solution. We prove that iteratively refined IF-QIPMs (IR-IF-QIPMs) have exponentially improved complexity w.r.t precision compared to previous QIPMs.

The last challenge in QIPMs is that QLSAs are sensitive to the condition number of linear systems arising in QIPMs and Newton systems are usually ill-conditioned. There are two major sources of ill-conditioning in QIPMs. First, for degenerate LO problems, the sequence of coefficient matrices of Newton systems converge to a singular matrix. i.e., their condition number grows to infinity. We show that a properly adapted iterative regiment technique helps with issues as we stop QIPMs early when the condition number is comparatively small enough. Another source of ill-conditioning is the ill-conditioned input matrix $A$. We address this issue by preconditioning the Newton system. In addition, we show how this particular preconditioner can be applied on a quantum machine without excessive cost [26].

# 4   Improved QLSA+QTA Subroutine for QIPMs

The idea of using iterative refinement for quantum algorithms is further used to develop improved QLSA+QTA subroutine for QIPMs [30]. The most efficient QLSA to solve a linear system of the form $Mz = \sigma$ with $\mathcal{O}(\log(\frac{p}{\epsilon})\kappa\|M\|_F)$ inquiries to QRAM [31], where $p$ is the system dimension, representing an exponential speedup over classical algorithms. A major hurdle lies in the fact that QLSAs solve Quantum Linear System Problems (QLSPs) and so the result is a quantum state, which deviates from the classical definition of the solution of LSPs. Consequently, a Quantum Tomography Algorithm (QTA) is essential to extract a classical solution. The best time complexity of QTA is $\mathcal{O}(\frac{p\varrho}{\epsilon})$, where $\varrho$ represents the upper bound on the norm of the solution.

The overall complexity of QLSA and QTA combined is $\mathcal{O}(\log(\frac{p}{\epsilon})\frac{p\kappa^2\|\sigma\|}{\epsilon})$. In comparison to the conjugate gradient method (CGM), its query complexity exhibits a better dependence on sparsity with unfavorable dependence on precision and condition number. An iterative classical-quantum linear system algorithm (ICQLSA) has been proposed which exponentially improves the time complexity of Quantum Linear Solvers, providing a classical solution with high precision up to $\mathcal{O}(\log(\frac{p\|\sigma\|}{\epsilon})p\kappa^2)$ queries to QRAM [30].

This new advancement enables us to do the calculations in high precision settings where $\epsilon = 2^{-2L}$ which is almost exact for the solution of the LO problems.

Thus we can also do matrix-vector multiplications on the quantum machine. Using ICQLSA and Quantum mat-vec product within the IR-IF-QIPM leads to optimal scaling $\mathcal{O}(n^2 \kappa_A L)$ as the worst case complexity for solving LO can not have better dimension dependence than quadratic as storing and reading dense matrix $A$ need $\mathcal{O}(n^2)$ arithmetic operations.

## 5   The state-of-the-art QIPM

In this section, we develop an almost exact quantum interior point method for solving linear optimization problems. Assuming that the input data is all integer, we denote the binary length of the input data by

$$L = mn + m + n + \sum_{i,j} \lceil \log_2(|a_{ij}| + 1) \rceil$$
$$+ \sum_i \lceil \log_2(|c_i| + 1) \rceil + \sum_j \lceil \log_2(|b_j| + 1) \rceil,$$

where $a_{ij}$ represents the $ij$-element of matrix $A$. The optimal partition is also defined as

$$\mathcal{B} = \{j \in \{1, \ldots, n\} : x_j^* > 0 \text{ for some } (x^*, y^*, s^*) \in \mathcal{PD}^*\},$$
$$\mathcal{N} = \{j \in \{1, \ldots, n\} : s_j^* > 0 \text{ for some } (x^*, y^*, s^*) \in \mathcal{PD}^*\}.$$

The following lemma is a classical result first proved by [32].

**Lemma 1.** *Let $(x^*, y^*, s^*) \in \mathcal{PD}^*$ be a basic solution. If $x_i^* > 0$, then we have $x_i^* \geq 2^{-L}$. If $s_i^* > 0$, then we have $s_i^* \geq 2^{-L}$.*

Lemma 1 is a fundamental result in the complexity analysis of IPMs. It means that after enough number of iterations of IPMs, a decision variable can be rounded to zero if it is smaller than $2^{-L}$. Then, by a rounding procedure, one can find an exact optimal solution for linear optimization [1,33]. In the proposed algorithm, all calculations happen on a quantum machine with precision $\epsilon = 2^{-tL}$ where $t$ is a small constant, less than 10. This high level of accuracy justifies describing the algorithm as *almost-exact*. The only calculation that happens on a classical computer is updating the solution and vector-vector summation. In this paper, we use the dual logarithmic barrier method, which has a simple framework. At each step of the dual log barrier IPM, we need to solve the following Newton system

$$\begin{bmatrix} I & A^T \\ AS^{-2} & 0 \end{bmatrix} \begin{bmatrix} \Delta s \\ \Delta y \end{bmatrix} = \begin{bmatrix} 0 \\ \frac{1}{\mu}(b - AS^{-1}e) \end{bmatrix}, \tag{1}$$

where $S = \text{diag}(s)$. Let $\hat{\Delta}s = S^{-2} \Delta s$, we can have the system

$$\begin{bmatrix} S^2 & A^T \\ A & 0 \end{bmatrix} \begin{bmatrix} \hat{\Delta}s \\ \Delta y \end{bmatrix} = \begin{bmatrix} 0 \\ \frac{1}{\mu}(b - AS^{-1}e) \end{bmatrix}. \tag{2}$$

One can easily verify that $M = \begin{bmatrix} S^2 & A^T \\ A & 0 \end{bmatrix}$ is a symmetric positive definite matrix, and so the system (2) has a unique solution [1]. Given $s$, one can build block-encodings of implementing matrix $M$ and preparing state $\sigma = \begin{bmatrix} 0 \\ \frac{1}{\mu}(b - AS^{-1}e) \end{bmatrix}$ efficiently, assuming that matrix $A$ stored in QRAM in advance. The general steps of the proposed almost exact QIPM using a short-step framework are described in Algorithm 1.

---

**Algorithm 1** Almost Exact QIPM

---

**INPUT** Dual feasible solution $(y^0, s^0)$, $\mu^0 > 0$, $0 < \theta < 1$, and $\delta\left((y^0, s^0), \mu^0\right) < \frac{1}{2}$, where $\delta$ is the proximity measure from [1, 28]

Store $A, b, c$ on QRAM

$k \leftarrow 1$

**while** $\mu > 2^{-2L}$ **do**

$\quad (\Delta y^k, \hat{\Delta s}^k) \leftarrow$ Solve system 2 with precision $\epsilon = 2^{-tL}$

$\quad y^{k+1} \leftarrow y^k + \Delta y^k$

$\quad s^{k+1} \leftarrow s^k + (S^k)^2 \hat{\Delta s}^k$

$\quad \mu^{k+1} = (1 - \theta)\mu^k$

$\quad k \leftarrow k + 1$

**end while**

---

As we analyze the worst-case complexity, we assume $m = \mathcal{O}(n)$ and matrices are fully dense.

**Theorem 1.** *Number of iterations for Algorithm 1 has upper bound*

$$\mathcal{O}(\sqrt{n}L).$$

We prove the theorem in the next section.

**Proof of Theorem 1** Suppose we start with a strictly feasible solution $(x_0, y_0, s_0)$. In the dual logarithmic barrier IPM, we do not compute the value of $x$ and $y$ but they exist. We have

$$Ax_0 = b, \ A^T y_0 + s_0 = c, \ s_0 > 0.$$

Then we use a quantum subroutine to compute an inexact $\Delta s_0$ with associated error $\xi_1$. After a full Newton step, we have

$$Ax_1 = b, \ A^T y_1 + s_1 = c + \xi_1, \ s_1 > 0.$$

Now we get a feasible solution for problem $(A, b, c + \xi_1)$. We do another full Newton step, then we have

$$Ax_2 = b, \ A^T y_2 + s_2 = c + \xi_1 + \xi_2, \ s_2 > 0.$$

We can keep doing this until we have

$$Ax_k = b, \ A^T y_k + s_k = c + \sum_{i=1}^{k} \xi_i, \ s_k > 0.$$

Then, we can rewrite all of them into

$$Ax_j = b, \ A^T y_j + s_j + \sum_{j+1 \leq k}^{k} \xi_l = c + r^k, \ s_j > 0,$$

where $r^k = \sum_{i=1}^{k} \xi_i$. This implies we obtained a series of feasible iterates for problem $(A, b, c+r^k)$. When their Newton steps are obtained exactly for problem $(A, b, c + r^k)$, then this series converges to an optimal solution for the problem in $\mathcal{O}(\sqrt{n})$ iterations. However, if their Newton steps are inexact but satisfy the conditions in [28], the $\sqrt{n}$ complexity still holds. But these Newton steps are artificial steps because we do not know exactly the errors $\xi_i$. We need to show the actual Newton steps we inexactly compute are close enough to these artificial Newton steps and the inexactness is acceptable for the convergence conditions.

In the first iteration, the actual and artificial Newton steps are computed as

$$\Delta s_0 = -A^T \left( AS_0^{-2} A^T \right)^{-1} \frac{1}{\mu_0} \left( b - \mu_0 AS_0^{-1} e \right) + \xi_1$$

$$\Delta \tilde{s}_0 = -A^T \left( A\tilde{S}_0^{-2} A^T \right)^{-1} \frac{1}{\mu_0} \left( b - \mu_0 A\tilde{S}_0^{-1} e \right),$$

where

$$\tilde{S}_0 = S_0 + r^k.$$

According to [28], we need

$$\left\| \tilde{S}_0^{-1} (\Delta s_0 - \Delta \tilde{s}_0) \right\|_2 \leq 0.1 \delta_{\tilde{c}} (\tilde{s}_0, \mu_0),$$

where $\delta_{\tilde{c}}$ is the proximity measure for the perturbed problem $(A, b, \tilde{c})$ with $\tilde{c} = c + r^k$. This condition can be guaranteed when

$$\left\| (S_0 \tilde{S}_0^{-1} (I - S_0 \tilde{S}_0^{-1})) \right\|_2 \leq 0.033 \delta_{\tilde{c}} (\tilde{s}_0, \mu_0),$$

$$\left\| I - (S_0 \tilde{S}_0^{-1})^2 \right\|_2 \leq 0.033, \tag{3}$$

$$\left\| \tilde{S}_0^{-1} \xi_1 \right\|_2 \leq 0.033 \delta_{\tilde{c}} (\tilde{s}_0, \mu_0).$$

Notice that all three conditions can be satisfied by pushing $\xi_i$ to be small as long as $\delta_{\tilde{c}}$ is not zero, which can be inferred by the approximate value of $\delta$. We discuss the value of $\xi_i$ later in the section. This proves that our inexact Newton step is a feasible inexact Newton step for the perturbed problem. Then, according to Theorem 3.3 of [28], we have the $\mathcal{O}(\sqrt{n}L)$ complexity.

After $\mathcal{O}(\sqrt{n}L)$ iterations, we have an $\tilde{x} > 0$ such that

$$A\tilde{x} = b,$$
$$A^T y^k + s^k = c + r^k,$$
$$(\tilde{x})^T s^k \leq 2^{-2L},$$

where $r^k = \sum_{i=1}^{k} \xi_i$. It is easy to verify that $(\tilde{x}, y^k, s^k)$ is a $2^{-tL}$-optimal solution for the perturbed problem $(A, b, c + r^k)$, and one can calculate the exact optimal solution by a rounding procedure. It is easy to verify that $\|r^k\| \leq 2^{(1-t)L}$. In the remaining, we show how we can retrieve an optimal solution of the original problem with a rounding procedure from the optimal solution for the perturbed problem.

It is straightforward to see that $(\tilde{x}, y^k, s^k)$ is in a $2^{(1-t)L}$-neighborhood of the optimal set for the original problem $(A, b, c)$. As the smallest nonzero element of $s^*$ and $x^*$ is greater than $2^{-L}$, using partitions $B$ and $N$ of this solution, by solving a constrained least squares problem, an optimal solution for the original problem can be obtained. For the details of the rounding procedures, refer to Chapter 7 of [33].

It is worth noting that the rounding procedures are strongly polynomial-time methods. They can also be quantized using quantum linear system solvers; however, we do not explore the cost and implementation details of rounding procedures in this paper, as it is beyond the scope of this paper.

### 5.1   Quantum Subroutine

In this section, we analyze the complexity of building and solving system (2). We use the general scheme of the Quantum Tomography framework of [30, 34]. We assume that we have access to a large enough QRAM, and we store data $A, b, c$ initially on QRAM with worst-case $\mathcal{O}(n^2)$ complexity. At each state we need to store $s$ on QRAM and build and solve System (2) using the iterative quantum linear solver of [34]. At each iteration of Algorithm 2, the only classical operation is updating the solution by a vector summation with $\mathcal{O}(n)$ arithmetic operations. In the following, we calculate the cost of quantum operations.

**Lemma 2.** *Given $A$ and $S$ stored on QRAM, the following statements are true:*

- *We can construct a block-encoding of $M$ using $\mathcal{O}(polylog(\frac{n}{\epsilon}))$ queries to QRAM.*
- *We can prepare the the state $|r\rangle$ using $\mathcal{O}(polylog(\frac{n}{\epsilon}))$ queries to QRAM.*
- *We can apply $M^{-1}$ using $\tilde{\mathcal{O}}_{n,\kappa,\frac{1}{\epsilon}}(\kappa\|A\|_F)$ queries to QRAM.* [1]
- *Norm estimation of $p^k$ and $r^k$ costs $\tilde{\mathcal{O}}_{n,\kappa,\frac{1}{\epsilon}}(\kappa\|A\|_F)$ queries to QRAM.*

The proof of Lemma 2 is the direct result of Prepositions 1 to 6 of [24].

---

[1] The $\widetilde{\mathcal{O}}_{\alpha,\beta}(g(x))$ notation indicates that quantities polylogarithmic in $\alpha, \beta$ and $g(x)$ are suppressed.

---

**Algorithm 2** Quantum Linear Solver

---

    **INPUT** $(A, b, c)$ stored on QRAM,

    Store $s$ on QRAM

    $k \leftarrow 1$

    $z^k \leftarrow 0$

    **while** $\|\Delta y^k - \Delta y^{k-1}\| > 2^{-4L}$ **do**

      Prepare State $|r^k\rangle = |\sigma - Mz^k\rangle$

      Apply inverse of block encoding of $M$ using QSVT [31]

      Extract classical solution $\frac{p^k}{\|p^k\|} = \frac{M^{-1}r^k}{\|M^{-1}r^k\|}$ via Tomography [35] with precision $\epsilon = 10^{-2}$

      Estimate norm of $\|p^k\|$ and $\|r^k\|$

      $z^{k+1} \leftarrow z^k + \frac{p^k}{\|r^k\|}$

      $k \leftarrow k + 1$

    **end while**

---

**Lemma 3.** *The number of iterations of Algorithm 2 is at most $\mathcal{O}(L)$.*

The proof of Lemma 3 is based on [34]. Additionally, the total complexity of Algorithm 2 is based on the analysis provided by [34].

**Theorem 2.** *Assuming $(A, b, c)$ is stored on QRAM, Algorithm 2 can find a $2^{-tL}$-precision solution for System (2) with*

$$\tilde{\mathcal{O}}_{n\kappa L}(n\kappa\|A\|_F)$$

*iterations.*

### 5.2 Proposed IR-AE-QIPM

In this section, we discuss how to use the iterative refinement method (IR) for LO problems to improve complexity as in [28] and provide the full description of our proposed algorithm. The first iterative refinement for LO has been proposed by [36]. Mohammadisiahroudi et al. [25] first showed that using iterative refinement can improve the complexity of QIPMs w.r.t precision and condition number. Further, in [29], the quadratically convergent iterative refinement scheme was proposed for feasible IPMs. An IR for dual log-barrier QIPM has been developed in [28].

In [28], the iterative refinement method for the LO problem works as follows:

1. Start with the original problem and solve it to a low accuracy;
2. If the accuracy of the original problem is not enough, construct a refining problem using the current iteration values; otherwise, the algorithm halts;
3. Solve the refining problem to a low accuracy and update the solution to the original problem; then, go to step 2.

In our proposed algorithm, after each solve, we have a feasible solution to a perturbed problem. To use the iterative refinement method, we need to construct

a solution to the original problem from the solution to a perturbed problem. To do so, we need a projection procedure. We use Algorithm 2 to solve the following problem

$$\min_{y} \|A^T y + s_k - c\|_2,$$

which is equivalent to

$$AA^T y = A(c - s_k).$$

Then we have

$$s = c - A^T y.$$

According to the argument in the previous section, this $(y, s)$ is feasible for the original problem with a duality gap bounded by twice of the low accuracy. Then, we can use the IR to refine the solution to high accuracy as in [28].

To get the full complexity of the proposed algorithm, we discuss the accuracy needed for $\xi_i$. In the first iteration, we need conditions (3). Theoretically, $\Delta_{\tilde{c}}$ might be zero, which implies the corresponding Newton system right-hand side is zero. We do not need to solve such Newton systems. Instead, we check the norm of the right-hand side vector. If the norm is too small ($\leq 2^{-4L}$), we update $\mu$ without computing the Newton step. Then, conditions (3) can be guaranteed when

$$\|\xi_i\|_2 \leq \mathrm{poly}\left(\frac{2^{-4L}}{n\kappa_{AS_0^{-1}}}\right) \approx \mathrm{poly}(2^{-4L}), \ \forall i \in [k].$$

This bound also works for the remaining iterations. Now, we present the pseudocode of our proposed algorithm and the main theorem.

---

**Algorithm 3** Iteratively Refined Almost Exact QIPM

---

**INPUT** Dual feasible solution $(y^0, s^0)$, $\mu^0 > 0$, $0 < \theta < 1$, $\delta\left((y^0, s^0), \mu^0\right) < \frac{1}{2}$, $\nabla^{(0)} = 1$, $0 < \zeta \ll \tilde{\zeta}$

Store $A, b, c$ on QRAM

$k \leftarrow 1$

$(y_1, s_1) \leftarrow$ Solve dual problem with accuracy $\tilde{\zeta}$

**while** $\nabla^{(k-1)} < \frac{1}{\zeta}$ **do**

    $\nabla^{(k)} \leftarrow \nabla^{(k-1)} \times \frac{1}{\zeta}$

    Construct the IR problem as in [28]

    $(\hat{y}, \hat{s}) \leftarrow$ Solve IR problem with accuracy $\tilde{\zeta}$ and project into proper subspace

    $y^{k+1} \leftarrow y^k + \frac{1}{\nabla^{(k)}} \hat{y}$

    $s^{k+1} \leftarrow c - A^T y^{(k)}$

    $k \leftarrow k + 1$

**end while**

---

**Theorem 3 (Lemma 13 of [28]).** *Algorithm 3 terminates after $\mathcal{O}(\frac{\log(\zeta)}{\log(\hat{\zeta})})$ iterations.*

For our purpose, we use $\zeta = 2^{-tL}$ and $\hat{\zeta}$ is constant. Thus the outer iteration has iteration bound $\mathcal{O}(L)$. We also have a matrix-vector product at each step of this IR scheme with cost $\mathcal{O}(n^2)$ arithmetic operations.

The major challenge in AE-QIPM Algorithm 1 is that the complexity of the quantum solver depends on the condition number, and the condition number grows in each iteration of AE-QIPM. As in IR Algorithm 3, we stop AE-QIPM early at fixed precision. It has been shown that with early termination $\kappa^{(k)} = \mathcal{O}(\kappa_0)$ where $\kappa_0$ is the condition number of the coefficient matrix for $(y^0, s^0)$ and it is constant [28].

### 5.3   Total Complexity

In this section, we put together all the elements discussed in the previous sections to calculate the total worst-case complexity of IR-AE-QIPM.

**Theorem 4.** *Algorithm 3 produces a $2^{(1-t)L}$ precise optimal solution of the LO problem using at most*

$$\tilde{\mathcal{O}}_{\kappa_0, n, \|A\|_F}(n^{1.5} L \kappa_0))$$

*queries to QRAM and $\mathcal{O}(n^2 L)$ classical arithmetic operations.*

*Proof.* The number of iterations of IR is bounded by $\mathcal{O}(L)$ based on Theorem 3. At each iteration we have $\mathcal{O}(n^2)$ cost of a classical matrix-vector product and the cost of AE-QIPM to solve the refining problem. Additionally to address $\|A\|_F$ in the complexity, one can initially normalize data by $\|A\|_F$, and inconsequence final precision should be increased by $\|A\|_F$ which appears in polylog. The quantum complexity is $\tilde{\mathcal{O}}_{n, L, \|A\|_F}(n^{1.5} \kappa_0 L)$ queries to QRAM, and $\mathcal{O}(nL)$ arithmetic operations at each step of AE-QIPM. Thus the total queries to QRAM is

$$\tilde{\mathcal{O}}_{\kappa_0, n}(n^{1.5} L \kappa_0)),$$

and the total number of classical arithmetic operations is bounded by $\mathcal{O}(n^2 L)$.

Table 1 compares the complexity of the proposed IR-AE-QIPM with other classical and quantum IPMs. As we can see, the total complexity of our approaches outperforms previous complexities. In the last line of the table, we show the complexity of the classical counterpart of the IR-AE-IPM using CG to solve the system. As we can see, the total complexity can not be better than $n^{2.5}$ in the classical version. This exhibits a clear quantum advantage compared to other algorithms in the literature. It should be mentioned that the quantum complexity of all QIPMs is the number of queries to QRAM. Without QRAM assumptions, some overheads may appear in complexities, although the quantum central path method of [37] is QRAM-free.

## 6   Applications in AI and Machine Learning

The integration of QLSAs and QIPMs has shown promising potential to accelerate core problems in machine learning. This section highlights the key applications that can benefit from these quantum techniques, as demonstrated in recent studies including [27, 39].

**Table 1.** Worst-case Complexity of different IPMs for LO

| Algorithm | Linear System Solver | Quantum Complexity | Classical Complexity |
|---|---|---|---|
| IPM with Partial Updates [1] | Low rank updates | | $\mathcal{O}(n^3 L)$ |
| Feasible IPM [1] | Cholesky | | $\mathcal{O}(n^{3.5} L)$ |
| II-IPM [7] | PCG | | $\mathcal{O}(n^5 L \bar{\chi}^2)$ |
| Robust IPM [8] | Fast Mat-Mul and Partial Update | | $\mathcal{O}(n^w L)$ |
| Quantum Central Path [37] | Hamiltonian Evolution | $\tilde{\mathcal{O}}(n^{3.5}\frac{\omega}{\epsilon})$ | |
| IR-IF-IPM [26] | PCG | | $\tilde{\mathcal{O}}_{\mu^0}(n^{3.5}L\bar{\chi}^2)$ |
| IR-IF-QIPM [29] | QLSA+QTA | $\tilde{\mathcal{O}}_{n,\kappa_A,\|A\|,\|b\|,\mu^0}(n^{1.5}L\kappa_A^2\omega^5)$ | $\tilde{\mathcal{O}}_{\mu^0}(n^{2.5}L)$ |
| IR-IF-QIPM [26] | Precond+QLSA+QTA | $\widetilde{\mathcal{O}}_{n,\|A\|_F,\frac{1}{\epsilon}}(n^{1.5}L\bar{\chi}^2)$ | $\tilde{\mathcal{O}}_{\mu^0}(n^{2.5}L)$ |
| IPM with approximate Newton steps [38] | Q-spectral Approx. | $\tilde{\mathcal{O}}_{n,\frac{1}{\zeta}}(n^{5.5})$ | $\tilde{\mathcal{O}}_{\frac{1}{\zeta}}(n^{1.5})$ |
| Quantum Dual-log Barrier [28] | QLSA+QTA | $\tilde{\mathcal{O}}_{n,\kappa_0,\mu^0,\|A\|_F}(n^{1.5}\kappa_0 L)$ | $\mathcal{O}(n^{2.5}L)$ |
| Proposed IR-AE-QIPM | IQLSA+Quant Mat-Vec | $\tilde{\mathcal{O}}_{n,\kappa_0,\|A\|_F}(n^{1.5}L\kappa_0)$ | $\mathcal{O}(n^2 L)$ |
| Classical IR-AE-IPM | CGM | | $\mathcal{O}(n^{2.5}L\kappa_0)$ |

Quantum-enhanced regression is one of the most direct applications of QLSAs in machine learning. Ordinary Least Squares (OLS), Weighted Least Squares (WLS), and Generalized Least Squares (GLS) problems can all be reduced to solving linear systems of the form $(X^T X)\beta = X^T y$, which QLSAs can handle efficiently. When paired with quantum tomography algorithms (QTAs), these solvers can retrieve classical solutions for model training and inference. The incorporation of iterative refinement techniques further enables exponential speedups with respect to precision, overcoming the classical bottleneck caused by ill-conditioning. Specifically, recQLSAs offer:

- Exponential speedup w.r.t. dimension in state preparation.
- Exponential speedup w.r.t. precision via iterative refinement.
- Milder dependence on condition number through adaptive regularization.

Many sophisticated machine learning models, such as Support Vector Machines (SVMs) and Lasso Regression, can be formulated as Linearly Constrained Quadratic Optimization (LCQO) problems. These problems are ideal candidates for QIPMs, which leverage QLSAs to solve Newton systems arising in Interior Point Methods. [27] proposed an Inexact Feasible QIPM (IF-QIPM) that preserves feasibility of iterates using orthogonal subspace systems (OSS), enabling the solution of LCQO problems including:

- Lasso Regression: Promotes sparse solutions using $\ell_1$ regularization. Reformulated as an LCQO problem, it can be solved with improved complexity using IF-QIPMs.
- Soft-Margin Support Vector Machines: Reformulated as LCQO using variable splitting and slack variables. QIPMs achieve better complexity in high-dimensional regimes.

These quantum-enhanced formulations demonstrate:

- Polynomial speedup w.r.t. dimension $n$ over classical IPMs.

– Exponential speedup w.r.t. precision and condition number over previous QIPMs.
– Improved feasibility guarantees through OSS-based feasibility maintenance.

Together, these applications showcase the growing relevance of QLSAs and QIPMs in machine learning, particularly as hardware capabilities advance. The combination of quantum speedups in dimension, precision, and matrix conditioning illustrates a compelling path forward for quantum-enhanced data science.

## 7   Conclusions

In this work, we presented recent advances in the development of Quantum Interior Point Methods (QIPMs) for Linear Optimization. By integrating iterative refinement and preconditioning techniques, we tackled two major challenges inherent in QLSA-based QIPMs: the inexactness of quantum solvers and their sensitivity to the condition number of the Newton system. We further introduced a novel Almost-Exact Interior Point Method framework, in which all matrix-vector operations and Newton system computations are performed on a quantum computer. This approach delivers a provable quantum speedup over classical IPMs.

To achieve exponentially small error in the computed Newton steps, we embed iterative refinement both internally within the quantum solver and externally across IPM iterations. As a result, the overall algorithm attains an optimal worst-case complexity of $\mathcal{O}(n^2)$ for fully dense linear optimization problems.

A key limitation of the proposed method is its dependence on Quantum Random Access Memory (QRAM), the physical implementation of which remains an open challenge. However, alternative approaches can be explored to mitigate this dependency. For example, circuit-based QRAM constructions [40], recent developments in Quantum Singular Value Transformation (QSVT) without block encoding [41], or sparse-access input models offer promising directions for developing QRAM-free variants of the algorithm. Moreover, a detailed resource estimation study, such as the framework in [42], is essential for evaluating the real-world feasibility and quantum advantage of the proposed method.

Future research could also focus on extending this framework to a primal-dual Almost-Exact QIPM applicable to both linear and semidefinite optimization problems. Primal-dual methods, particularly those based on self-dual embedding formulations, offer the advantage of not requiring an initial strictly feasible interior point, thus expanding the applicability of QIPMs in practice.

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
