# OpenReview forum: "Quantum Interior Point Methods: A Review of Developments and an Optimally Scaling Framework"
_purdue.edu/Purdue_University/PQAI/2025/Symposium — PQAI 2025 Oral_

### Official Review · Reviewer_Amqz · 2025-07-25
**Clear and informative contribution**

**Rating:** 8
**Confidence:** 3

**Review:**

The manuscript “Quantum Interior Point Methods: A Review of Developments and an Optimally Scaling Framework” by Mohammadisiahroudi et al. is a review of recent developments of Quantum Interior Point Methods (QIMPs) for treating linear optimisation problems. As the authors rightly highlight, quantum linear systems are among the most interesting and promising classes of problems to be address by quantum computing, The work is structured as a review, still keeping a good technical level. In the last sections, applications in AI and ML, as well as an outlook for future research directions are provided.

I believe that the goal of the authors is a review submission; the manuscript aligns well to this goal. The treatment is only slightly biased towards the work connected to the authors, but still within reasonable bounds. To me as a reader there was an unclear point in Section 4, when the authors state “Thus we can also do matrix-vector multiplications on the quantum machine.” because I do not find the link between this statement and the previous part of the section. Otherwise, this is an interesting and informative work.

---

### Decision · Program_Chairs · 2025-07-29

Accept (Oral)